# Unveiling the Enigmatic Adenoids and Tonsils: Exploring Immunology, Physiology, Microbiome Dynamics, and the Transformative Power of Surgery

**DOI:** 10.3390/microorganisms11071624

**Published:** 2023-06-21

**Authors:** Pinelopi Samara, Michael Athanasopoulos, Ioannis Athanasopoulos

**Affiliations:** 1Children’s Oncology Unit “Marianna V. Vardinoyannis-ELPIDA”, Aghia Sophia Children’s Hospital, 11527 Athens, Greece; 2Otolaryngology-Head & Neck Surgery, Athens Pediatric Center, 15125 Athens, Greece; miathanasopoulos@gmail.com (M.A.); athanasopoulosj@hotmail.com (I.A.)

**Keywords:** adenoids, tonsils, Waldeyer’s ring, microbiome, antibiotics, adenoidectomy, tonsillectomy, adenotonsillectomy

## Abstract

Within the intricate realm of the mucosal immune system resides a captivating duo: the adenoids (or pharyngeal tonsils) and the tonsils (including palatine, tubal, and lingual variations), which harmoniously form the Waldeyer’s ring. As they are strategically positioned at the crossroads of the respiratory and gastrointestinal systems, these exceptional structures fulfill a vital purpose. They function as formidable “gatekeepers” by screening microorganisms—both bacteria and viruses—with the mission to vanquish local pathogens via antibody production. However, under specific circumstances, their function can take an unsettling turn, inadvertently transforming them into reservoirs for pathogen incubation. In this review, we embark on a fascinating journey to illuminate the distinctive role of these entities, focusing on the local immune system inside their tissues. We delve into their behavior during inflammation processes, meticulously scrutinize the indications for surgical intervention, and investigate the metamorphosis of their microbiota in healthy and diseased states. We explore the alterations that occur prior to and following procedures like adenoidectomy, tonsillectomy, or their combined counterparts, particularly in pediatric patients. By comprehending a wealth of data, we may unlock the key to the enhanced management of patients with otorhinolaryngological disorders. Empowered with this knowledge, we can embrace improved therapeutic approaches and targeted interventions/surgeries guided by evidence-based guidelines and indications.

## 1. Introduction

Waldeyer’s ring, named after the distinguished German anatomist, Heinrich Wilhelm Gottfried von Waldeyer-Hartz (6 October 1836–23 January 1921), is a remarkable collection of lymphoid tissues that spans the naso- and oropharynx [1,2]. It appears during the fifth month of gestation and consists of four distinct structures: (1) pharyngeal tonsils or adenoids, located on the roof of the nasopharynx under the sphenoid bone; (2) tubal tonsils, found bilaterally surrounding the opening of the Eustachian tube in the lateral wall of the nasopharynx; (3) palatine tonsils, which are traditionally referred to as “the tonsils”, situated within the tonsillar clefts of the oropharynx; and (4) lingual tonsils, a group of lymphatic tissue characterized by numerous protrusions at the posterior third of the tongue [3,4] (Figure 1). Moreover, mucosa-associated lymphoid tissue (MALT) can be found within these tonsils and throughout the naso- and oropharynx [5,6]. All these entities form the mucosal immune system, which is strategically placed at the intersection between the respiratory and digestive systems [7]. They play a crucial role in defending against pathogens introduced through inhalation and digestion, serving as primary sites for antigen sampling and triggering essential immune responses [8,9].

However, sometimes, chronic inflammation caused by microbes, mainly bacteria, can lead to adenotonsillar disease [10]. Ιn these cases, an appropriate treatment must be administered, including proper antibiotics or, if indicated, the surgical removal of implicated structures that create “more harm than good” [11,12]. The accurate diagnosis and effective management of adenotonsillar infections highly depend on in-depth knowledge of the anatomy, physiology, immunology, microbiology, and pathophysiology of the area and its underlying tissues. In the past, there were two schools of thought concerning the necessity of these surgical interventions (adenoidectomy, tonsillectomy, or adenotonsillectomy) [13]. Nowadays, it is widely accepted that there are specific indications for their removal, with the primary objective of improving the patients’ quality of life. In fact, adenoidectomy and tonsillectomy are among the most commonly performed surgical procedures for the treatment of children worldwide [14].

The goal of this review is to serve as a mini-guide that includes up-to-date information concerning the tonsils and adenoids for both clinicians and laboratory practitioners. We analyze the unique role of Waldeyer’s ring and its constituent parts in the local immune system, exploring their microbiome and its potential alterations in healthy and diseased states before and after an adenoidectomy, tonsillectomy, or their combined procedures, particularly in children. Furthermore, we investigate their behavior during inflammatory processes. Ultimately, this manuscript strives to contribute to the improvement of treatment approaches and the development of more targeted strategies aligned with established guidelines and recommendations. 

## 2. Anatomy and Physiology of Adenoids and Tonsils

For pediatric otolaryngologists, the pharyngeal and palatine tonsils are of utmost importance in their daily clinical practice [15]. Therefore, we will place greater emphasis on these two structures and provide a more comprehensive description of them.

### 2.1. Anatomy of Pharyngeal Tonsils (Adenoids)

The pharyngeal tonsils, which are also called adenoids, are lymphoid tissue located in the central area where the sphenoid and occipital bones meet along the roof and posterior wall of the nasopharynx. Below the lower border of the pharyngeal tonsil, the pharyngobasilar fascia is present, which extends downward and is associated with the pharyngeal constrictor muscles [16]. Grisel’s syndrome, a rare complication that typically occurs after an adenoidectomy, can result from inflammation caused by aggressive adenoid removal and/or excessive backward bending in patients with certain anatomical predispositions [17]. The blood supply to the pharyngeal tonsils comes from various sources, including the ascending pharyngeal artery, ascending palatine artery, tonsillar branch of the facial artery, pharyngeal branch of the internal maxillary artery, and artery of the pterygoid canal [18].

### 2.2. Anatomy of Palatine Tonsils

The palatine tonsils are lymphoid structures located within the tonsillar fossa. This region is bordered by the anterior and posterior tonsil pillars, which consist of the palatoglossus and palatopharyngeus muscles [19]. The lateral side of the tonsillar fossa is formed of the superior constrictor muscle. The buccopharyngeal fascia and a layer of loose connective tissue separate the tonsil from the parapharyngeal space [4]. Blood supply to the tonsil is provided by branches of the external carotid artery system, including the tonsillar artery and the ascending palatine artery. The superior pole of the palatine tonsil receives blood from the tonsillar branches of the ascending pharyngeal artery and the descending palatine artery. During procedures involving the palatine tonsils, healthcare providers must be cautious of nearby neurovascular structures, such as the internal carotid artery and the glossopharyngeal nerve [20,21]. In adults, the internal carotid artery is typically located 2.5 cm posterolateral to the tonsillar fossa. However, in children weighing under 11 kg, it can be as close as 1.5 cm to the fossa [22].

### 2.3. Physiology of Adenoids and Tonsils

It is important to briefly discuss the embryological development of the palatine and pharyngeal tonsils in order to gain a better understanding of their physiology. The medial epithelial surface of the tonsil originates from the second branchial pouch, where solid epithelial cores penetrate into the surrounding mesenchyme. Over time, these cores undergo canalization and form small invaginations called crypts. At around the 16th–17th week of embryological development, lymphocytes and lymphoid stem cells invade the deepest layers of the connective tissue, forming follicles and germinal centers [23]. The growing lymphoid components merge the deepest layers of the connective tissue to create a thin tonsillar capsule. After birth, the tonsils develop multiple branching crypts, with approximately 10–30 per tonsil. These crypts have a fibrovascular core surrounded by lymphoid tissue. The tonsillar epithelial surface is composed of non-keratinized stratified squamous epithelium, while the lining of the crypts consists of stratified squamous epithelium and lymphoepithelium [24]. The invaginated structure of the crypts increases the surface area for antigen sampling and the direct trapping of foreign materials. Unlike lymph nodes and the spleen, the palatine tonsils lack lymphatic vessels for fluid drainage [25].

As for the pharyngeal tonsils, they exhibit mucosal folds, but have fewer crypts compared to those of palatine tonsils. Histologically, the pharyngeal tonsils are primarily composed of pseudostratified ciliated columnar epithelium with a smaller number of lymphoid follicles. A capsule separates the pharyngeal tonsil from the surrounding bones, and connective tissue septa divide the tissue into 4–6 segments [25].

## 3. Immunology of Adenoids and Tonsils: A Regionalized Immune System

The unique anatomical arrangement of adenoids and tonsils, combined with specialized immune cell populations, creates a regionalized immune system that helps prevent the spread of pathogens and elicits immune responses [26]. In this section, we will briefly present the structural organization of adenoids and tonsils, examine their immune cell populations, and delve into their functional significance in terms of mucosal immunity.

The epithelium covering these lymphoid structures contains specialized cells, such as M cells and intraepithelial lymphocytes, which contribute to the immune surveillance of luminal antigens. Below the epithelium, the lamina propria harbors various immune cell populations, including B cells, T cells, dendritic cells (DCs), and macrophages, forming an organized network of lymphoid follicles and interfollicular areas. B cells, which are abundant, undergo germinal center reactions within the adenoids and tonsils, leading to antibody production and affinity maturation. T cells, including CD4+ helper T cells and CD8+ cytotoxic T cells, play a critical role in coordinating immune responses and eliminating infected cells [27]. DCs that are strategically positioned at the interface between the lumen and lymphoid tissue capture and present antigens to T and B cells, initiating adaptive immune responses. Macrophages distributed throughout the lymphoid tissue phagocytose pathogens and provide important regulatory signals to other immune cells [8].

The regionalized immune system of the adenoids and tonsils serves several important functions. Firstly, it acts as a physical barrier, preventing pathogens from gaining access to the underlying tissues. The tonsillar crypts and M cells facilitate antigen uptake and transport, initiating immune responses. Secondly, adenoids and tonsils participate in the generation of adaptive immune responses. B cells within germinal centers undergo class switch recombination and somatic hypermutation, resulting in the production of high-affinity antibodies. T cells recognize and eliminate infected cells, thus limiting the dissemination of pathogens. Thirdly, adenoids and tonsils contribute to the development of immune memory, promoting faster and more efficient immune responses upon re-exposure to previously encountered pathogens [28,29]. Despite them being physically close and having similar tissue characteristics, a study conducted by Stanisce et al. [30] has revealed significant variations in the cellular composition of adenoids and tonsils, specifically in functionally important immune and stromal subsets. These differences hold significant implications in terms of immunology, pathology, and physiology. 

The immune system of these organs undergoes alterations following inflammation, which could contribute to the development of “local issues” like peritonsillar abscess, which is discussed further below. Specifically, chronic or recurrent tonsillitis disrupts the tonsillar immune system, resulting in the shedding of M cells and weakening of the immunologic response to antigens [31]. However, the exact role of the abnormal local immune response or immunological impairments as risk factors for peritonsillar abscess development remains unclear. Peritonsillar abscesses are rarely observed in patients with general immunodeficiency, but they have been associated with infectious mononucleosis. Epstein–Barr virus (EBV) infection impairs humoral immunity, potentially affecting the coating of bacteria with immunoglobulins on tonsillar tissue. It may also lead to a temporary decrease in T-cell-mediated immunity, increasing bacterial attachment and colonization in the tonsils. Two studies have investigated the immunological changes or impairments in peritonsillar abscess patients without concurrent EBV infection. The first study discovered that only a small portion of bacteria in the pus showed opsonization via immunoglobulin or complement components. This could be due to the rapid phagocytosis or encapsulation of the infection, preventing immune cells from effectively attacking the bacteria within. The second study showed that human beta-defensins, which possess antimicrobial properties, were also detected in the tonsillar surface and abscess, but not in the lymphatic follicles. However, their precise function remains largely unexplored [32]. Further investigations into the immunological aspects of adenoids and tonsils will provide valuable insights into the mechanisms underlying mucosal immune responses and may lead to the development of novel strategies for immune-based interventions.

## 4. Microbiology of Adenoids and Tonsils

While the immunological role of adenoids and tonsils has been extensively studied, understanding their microbiology is equally crucial. These lymphoid tissues harbor a diverse microbial community, collectively known as the microbiota, which interacts with the local immune system and contributes to the maintenance of mucosal homeostasis [33]. Investigating the microbiology of adenoids and tonsils can provide valuable insights into the complex interplay between microbes and the host immune system, influencing health and disease outcomes (Figure 2). 

### 4.1. The Normal Microbiome and the Microbiome in Adenotonsillar Disease

The colonization of the nasopharynx starts in infancy and develops over months [34]. The adenoids and tonsils host a diverse microbiome that contributes to achieving an immune balance and mucosal health. However, the composition of this microbiome can differ between individuals due to factors like age, environmental exposures, and overall health [35]. Commonly, potentially pathogenic bacteria can be found in the nasopharynx of healthy children, either as transient or regular components of the nasopharyngeal flora. These bacteria may include *Neisseria*, *Streptococcus pyogenes*, *Haemophilus influenzae*, *Staphylococcus aureus*, *Actinomyces*, *Bacteroides*, *Prevotella*, *Porphyromonas*, *Peptostreptococci*, and *Fusobacterium* species [36,37,38].

Recent studies have provided insights into the composition and function of the microbiome in adenotonsillar disease. These investigations employed various techniques, including in vitro cultures, 16S rRNA gene sequencing, and tissue sample analysis, to explore microbial communities [39,40]. Several studies utilizing cultivation and PCR techniques suggest that *Helicobacter pylori* exposure occurs during early childhood and may contribute to the development of chronic adenotonsillitis, particularly in regions with a high prevalence [41,42]. It is important to note that swab-based culture methods only identify bacteria present on the surface of the sampled tissue area, potentially missing those residing in deep crypts or intracellular regions. Consequently, culture swabs have limitations in clinical practice as they can only grow bacteria with specific metabolic requirements that align with the culture media and atmospheric conditions employed. By comparing the microbiomes of healthy individuals and those of patients with adenotonsillar disease, researchers aim to uncover crucial differences and understand the microbial factors involved in disease development. Importantly, the adenotonsillar microbiota’s composition may vary and have specific roles in both surface and core tissues [43].

In pediatric populations, the microbial composition has been observed to differ between patients with tonsillar hyperplasia and recurrent tonsillitis. Jensen et al. [44] reported variations in the tonsillar crypt microbiota based on age and health status. *Haemophilus influenzae*, *Neisseria* species, and *Streptococcus pneumoniae* are exclusively found in children, while *Prevotella*, *Actinomyces*, *Parvimonas*, *Veillonella*, and *Treponema* are more abundant in adults. Children with tonsillar hyperplasia exhibit a predominance of *Streptococcus*, *Neisseria*, *Prevotella*, *Haemophilus, Porphyromonas*, *Gemella*, and *Fusobacterium* species within the tonsillar crypts. On the other hand, *Fusobacterium necrophorum*, *Streptococcus intermedius*, and *Prevotella melaninogenica/histicola* are associated with recurrent tonsillitis in adults. Notably, significant differences in phylogenetic community structures were observed between healthy adults and those with recurrent tonsillitis, as well as between children with recurrent tonsillitis and those with tonsillar hyperplasia.

In a study by Kim and Min [45], the microbiomes of adenotonsillar tissues in pediatric patients who snore were analyzed. The tonsil tissue samples were primarily composed of *Proteobacteria* (mainly the *Haemophilus* genus), *Firmicutes* (predominantly the *Streptococcus* genus), and *Bacteroidetes* (dominated by the *Prevotella* genus). Adenoid tissue samples were dominated by *Proteobacteria* (mostly the *Haemophilus* genus), *Firmicutes* (mainly the *Streptococcus* genus), and *Fusobacteria* (predominantly the *Fusobacterium* genus). Furthermore, Swidsinski et al. [46] examined the tonsil and adenoid tissues and found that many patients had multiple areas of ongoing purulent infection within these tissues. The traditional notion that chronic adenotonsillitis is solely caused by a single bacterial species colonizing the tissue surface is being challenged [47]. Instead, tissue hyperplasia and enlargement may be influenced by a variety of opportunistic, commensal, and pathogenic microorganisms, as well as the immune system’s response to them. 

It would be an oversight to disregard the wide range of viruses that colonize the tonsils and adenoids. The list of respiratory viruses associated with upper airway infections is extensive and includes well-known agents, such as adenoviruses, bocavirus, coronaviruses, enteroviruses, EBV, human metapneumovirus, influenza viruses, parainfluenza viruses, respiratory syncytial virus, and rhinoviruses. Additionally, cytomegalovirus, human herpes viruses 6–8, herpes simplex virus, human papillomaviruses, human parvovirus B19, and polyomaviruses have been detected in the tonsillar and adenoidal tissues of asymptomatic children [48]. Remarkably, it has been reported that up to 97% of tonsils and adenoids harbor detectable viruses, and co-infections with multiple viral types have also been observed. While approximately 80% of adenoidal tissues may contain multiple viruses, slightly lower rates ranging between 59% and 68% have been reported in tonsillar tissues. These findings indicate that the presence of viruses in the tonsils and adenoids can be considered as a “normal viral flora”, suggesting that certain respiratory viruses may exhibit more chronic behavior [49]. It is worth noting that data from the COVID-19 pandemic have shown that tonsils and adenoids could serve as significant sites of SARS-CoV-2 infection in asymptomatic children [50].

In HIV patients, specific alterations in their tonsils and adenoids have been documented. Bacteria and yeast isolates, including *Staphylococcus aureus*, *Streptococcus pyogenes*, *Klebsiella pneumoniae*, *Escherichia coli*, *Proteus mirabilis*, *Candida albicans*, and *Candida tropicalis*, have been found in the respiratory tracts of HIV-positive children in Cambodia and Kenya. Additionally, a significant percentage of these children exhibited HIV-like sequences in these bacteria and yeasts, suggesting the potential for horizontal gene transfer between eukaryotic and prokaryotic cells, which could impact the progression of HIV disease [51].

All the aforementioned findings propose a potential connection between dysregulated microbiomes and the development or progression of adenotonsillar disease. Further research utilizing advanced sequencing techniques and metagenomic analyses is necessary to gain a comprehensive understanding of the adenoids and tonsils’ microbiomes and their potential implications in immune regulation and disease susceptibility.

### 4.2. The “Pathogen Reservoir” Hypothesis Relates to Biofilm Formation

The “Pathogen Reservoir” Hypothesis suggests that tonsils and adenoids serve as reservoirs for bacteria, contributing to recurrent upper respiratory tract infections. These lymphatic tissues can harbor microorganisms, leading to the formation of protective biofilms. Biofilms develop when bacteria adhere to surfaces and create a matrix that defends against the immune response and antimicrobial treatments. Several studies have highlighted the role of biofilms in chronic [52] and recurrent [53] tonsillitis, demonstrating their persistence and involvement in infection recurrence. Furthermore, biofilms contribute to antibiotic resistance by reducing the susceptibility to antimicrobial agents.

Bacterial biofilms also play a role in chronic adenoiditis and its associated complications, such as middle ear diseases [54]. Biofilms are present throughout the nasopharyngeal mucosa, particularly in the lateral region of adenoidal pads near the Eustachian tube, suggesting their involvement in chronic inflammation and middle ear problems [55]. Antibiotic therapy often fails in chronic adenoiditis cases due to biofilm resistance, which is facilitated by the physical barrier of the extracellular matrix and unique biofilm characteristics. Nasopharyngeal biofilms are associated with persistent or recurrent middle ear diseases, including chronic otitis media [56]. These biofilms impede antibiotic diffusion, exhibit reduced bacterial replication, and acquire resistance mechanisms. Young children with recurrent acute otitis media have a higher prevalence of nasopharyngeal biofilm-producing bacteria. Some studies challenge the assumption of sterile conditions in otitis media, as bacterial DNA has been found in middle ear effusion. Scanning electron microscopy reveals extensive biofilm coverage on adenoids in children with recurrent acute otitis media [57]. The specific location of biofilms near the Eustachian tube or tonsils further underscores their significance in various upper respiratory tract conditions.

These findings greatly contribute to our understanding of the “Pathogen Reservoir” Hypothesis and emphasize the importance of exploring strategies that target biofilms for the effective management of recurrent infections in the tonsils, adenoids, and ears.

## 5. Clinical Manifestations and Pathogenesis of Adenotonsillar Disease in Children and Adults

Recent research has enhanced our understanding of adenotonsillar disease, which presents differently in children and adults. Children commonly experience recurrent sore throat, difficulty swallowing, snoring, mouth breathing, and sleep disturbances. Chronic adenotonsillitis can affect their growth and development. In contrast, adults may have a chronic sore throat, persistent bad breath, voice changes, and discomfort while swallowing. Infections caused by *Streptococcus pyogenes*, *Haemophilus influenzae*, and EBV can trigger inflammation in the adenoids and tonsils, leading to their enlargement. Narrow upper airways or anatomical abnormalities also contribute to the development of adenotonsillar disease.

Adenotonsillar diseases encompass various categories, including tonsillitis, peritonsillar abscesses, and tonsil and adenoid hypertrophy. Tonsillitis, caused by bacterial or viral infections, refers to the inflammation of the tonsils, and it can be acute or chronic. Viral tonsillitis is typically associated with viruses causing the common cold [58]. Lemierre’s syndrome, a severe complication, can be caused by the anaerobic bacterium, *Fusobacterium necrophorum*, and can lead to emboli formation and septic infarctions [59]. Recurrent tonsillitis is commonly associated with *Staphylococcus aureus*, *Haemophilus influenzae*, *Haemophilus parainfluenzae*, and *beta-hemolytic streptococci*. In some cases, nonfermenter species like *Burkholderia cenocepacia* or *Enterobacteria* can be cultured from tonsil specimens. Persistent bacteria within biofilms are likely to be responsible for recurrent tonsillitis. *Staphylococcus aureus* strains associated with recurrent tonsillitis originate from the skin, nose, and pharynx, where they can form biofilms. Tonsillar hypertrophy is influenced by immunological factors and genetic predispositions. Dysfunctions in local lymphocytes and persistent inflammatory reactions contribute to the development of recurrent tonsillitis and tonsillar hypertrophy.

A peritonsillar abscess is a dangerous complication characterized by the formation of a pus-filled abscess between the tonsillar capsule and superior constrictor muscle, commonly involving *Streptococcus pyogenes* and anaerobic bacteria like *Fusobacterium*. This condition can lead to further health complications, posing a significant risk to the affected individual [60]. While most cases occur following acute tonsillitis [61], peritonsillar abscesses can also occur in post-tonsillectomy patients, possibly due to residual tonsillar tissue or fistula formation. Additionally, the obstruction of Weber glands, which are minor salivary glands, may contribute to the development of abscesses [62]. Differentiating between peritonsillar cellulitis and an abscess can be challenging due to them having similar symptoms. Cellulitis (or phlegmon) is characterized by erythematous and edematous tissue between the tonsil and its capsule, without obvious pus formation. 

The treatment options for peritonsillar abscess include needle aspiration, incision and drainage, antibiotics, supportive care, or a tonsillectomy. The frequency of immediate (à chaud or quinsy) tonsillectomies in current medical practice has decreased due to the greater difficulties posed by acute inflammation and the higher risk of hemorrhage. Hospitalization may be necessary for patients with complications or unsuccessful outpatient management. Corticosteroids have shown benefits in reducing pain and promoting faster recovery, although further research is needed to establish their routine use [61]. While the exact reasons for the development of uncomplicated tonsillitis versus peritonsillar abscesses remain unknown, certain risk factors have been identified. These include a median age of 20–39 years, having a history of recurrent tonsillitis, an extraperitonsillar spread on computerized tomography (CT), and receiving less than 3 days of intravenous antibiotic therapy. Furthermore, it has been observed that smokers are at an increased risk of developing a peritonsillar abscess [63,64]. Further research is needed to fully understand the underlying mechanisms and factors contributing to its development and recurrence. Hsiao et al. [65] conducted a study on peritonsillar abscesses in children, focusing on early recognition and appropriate management. *Streptococcus* species were the predominant organisms identified. In eight cases, the infection was effectively treated with antibiotics alone, while the remaining cases necessitated drainage procedures. Encouragingly, no fatalities were recorded. Two-thirds of the cases involved children aged 12 or older. In cases where patients were younger than 12, a CT scan was often necessary for a definitive diagnosis.

Adenoid hypertrophy involves the enlargement of the adenoids due to chronic inflammation or infections. The symptoms include nasal congestion, mouth breathing, and sleep disturbances. Bulfamante et al. [66] highlighted the association of adenoid hypertrophy with nasal symptoms in children with chronic rhinosinusitis. Enlarged adenoids obstruct the nasopharynx, leading to discomfort and reliance on mouth breathing. The quality of the voice may be affected, resulting in a twanging or nasal quality. Prolonged adenoid hypertrophy can cause an obstruction in the nasopharynx, leading to adenoid faces, sinusitis, otitis media, and middle ear effusions. Adenoids serve as reservoirs for pathogenic agents, contributing to recurrent infections.

## 6. Diagnosis and Management of Adenotonsillar Disease 

The diagnosis and management of adenotonsillar disease involve a thorough physical examination, including the evaluation of the tonsils, adenoids, uvula, and palate. Endoscopic nasopharyngoscopy is necessary for assessing the adenoids. The clinical presentation, history, and symptoms of the patient should also be considered. Additional tests may be required based on the findings. Radiographs and sleep tape recordings can be used to document snoring or apnea episodes if needed. Polysomnography is performed in high-risk children with a respiratory dysfunction. Optional sinus imaging may be conducted in pediatric patients undergoing adenoidectomy for chronic rhinosinusitis [67].

In addition to clinical evaluation, diagnostic methods and scoring systems, like Centor and McIsaac, play a valuable role in assessing A beta-hemolytic *streptococcus* (GABHS) infections, which are recognized as the most common bacterial cause of pharyngotonsillitis. These diagnostic tools are crucial in accurately identifying the presence of GABHS and guiding appropriate treatment strategies. The Centor score is suitable for patients aged 15 years and above, whereas the McIsaac score can be used in both adults and children. A score of three or higher on either criterion may suggest the need for further diagnostic tests, such as rapid tests or cultures. Rapid Antigen Detection Tests are effective in providing quick results (<10 min) by detecting streptococcal antigens [68]. Throat culture is a cost-effective diagnostic method known for its high sensitivity and specificity in identifying GABHS bacteria. This procedure involves obtaining a pharyngeal swab and culturing it to detect the presence of bacteria. To ensure that accurate and reliable results are obtained, the sampling technique employed during the pharyngeal swab is of utmost importance. It is critical to firmly press down the tongue and gently rub the swab in a turning motion across both tonsils or the lymphatic strands and the posterior pharyngeal wall. Care should be taken to avoid making any additional contact with the intraoral mucosa or saliva during the sampling process, as it may compromise the diagnostic quality of the sample [69]. Nucleic Acid Amplification Tests (NAATs) are molecular diagnostic techniques that amplify and detect specific nucleic acid sequences for GABHS identification. However, it is important to note that NAATs cannot differentiate between infection and carrier states [70]. Additionally, the integration of machine learning and artificial intelligence techniques can enhance the accuracy and efficiency of GABHS pharyngotonsillitis diagnosis via algorithm-based analysis [71,72].

### 6.1. Conservative Strategies-Antibiotic Treatment

Acute tonsillitis typically follows a self-limiting course, but the prompt administration of an appropriate antibiotic treatment is decisive, especially for addressing GABHS infections, as it offers several benefits. These benefits include reducing the duration of symptoms, minimizing transmission to close contacts, and preventing complications, such as acute rheumatic fever, scarlet fever, glomerulonephritis, peritonsillar abscess, etc. Therefore, it is recommended that patients with a positive rapid antigen detection test or throat culture, regardless of age, should receive antibiotics. Penicillin or amoxicillin are the preferred antibiotics for treating GABHS pharyngotonsillitis, and further analysis-based algorithms for antibiotic selection are described in the comprehensive review by Windfuhr and colleagues [58].

For symptomatic relief, particularly within the first three days of symptom onset, medications, like paracetamol (acetaminophen), and non-steroidal anti-inflammatory drugs, such as ibuprofen, can be used effectively. However, acetaminophen should be avoided if there is suspicion or confirmation of an EBV infection due to the risk of hepatotoxicity. The effectiveness of local anesthetics and antiseptics in the form of pharyngeal sprays, lozenges, and oral rinses has not been established. Patients should experience relief from symptoms within 48 h of initiating the treatment. If the symptoms persist, therapy compliance should be assessed, and a re-evaluation of the diagnosis should be considered.

### 6.2. Surgical Procedures: Adenoidectomy, Tonsillectomy, or Adenotonsillectomy and Potential Postoperative Complications

Specialized otolaryngologists should carefully evaluate specific indications before deciding to proceed with adenoidectomy and tonsillectomy surgeries [73]. It is crucial to thoroughly assess the potential risks and benefits associated with these interventions prior to making recommendations [74].

An adenoidectomy involves the excision of the adenoids, which are located in the back of the nasal cavity. This procedure is performed to alleviate symptoms, such as nasal obstruction, chronic sinusitis, recurrent ear infections, and sleep-disordered breathing (SDB). Various surgical instruments and techniques have been employed for adenoid removal, such as suction diathermy, curettage, microdebriders, electronic molecular resonance tools, endoscopy, and lasers [75,76,77,78]. Presently, otolaryngologists utilize endoscopic-assisted adenoidectomy with general anesthesia, followed by the utilization of a microdebrider to effectively scrape the adenoid tissue.

A tonsillectomy, on the other hand, involves the complete removal of the tonsils, along with their capsules. It can be performed independently or in conjunction with adenoidectomy. The indications for tonsillectomy include recurrent tonsillitis, peritonsillar abscess, obstructive SDB, and suspected malignancy. Various surgical techniques are employed for tonsillectomy, like cold steel dissection, electrocautery, cryosurgery, harmonic scalpel, laser tonsillectomy, bipolar and monopolar diathermy dissection, radiofrequency ablation, and coblation methods [79,80,81].

Following an adenoidectomy and a tonsillectomy, patients may experience several postoperative sequelae [82]. These include pharyngeal discomfort, nausea/vomiting, dehydration, a hemorrhage, and a fever [83]. Pharyngeal discomfort is a common and expected symptom, which can be managed with appropriate pain medication [84]. Nausea/vomiting may occur due to the effects of anesthesia or swallowing difficulties in the immediate postoperative period. It is important to provide antiemetic medications and ensure adequate hydration to minimize these symptoms and prevent dehydration. Hemorrhaging within the oral cavity is a potential complication that can occur. Immediate bleeding, which is often characterized by bright red blood, should be promptly evaluated and managed to control the bleeding source. In some cases, surgical intervention or cauterization may be required to achieve hemostasis. Delayed bleeding, which can occur several days after the surgery, may present with bleeding accompanied by darker-colored blood or clots. Patients should be advised to seek medical attention if bleeding occurs [85]. A fever is another possible postoperative complication. While a mild increase in temperature is common in the first few days after surgery, a persistent or high-grade fever may indicate an infection [86]. Infections can occur at the surgical site or in the surrounding areas and appropriate antibiotic treatment may be necessary for their management.

It is crucial to manage these complications promptly and provide appropriate postoperative care to ensure optimal recovery and patient comfort. Close monitoring, adequate pain management, hydration, and educating the patient about the signs of complications are essential components of postoperative care following an adenoidectomy and a tonsillectomy.

### 6.3. Indications for Adenoidectomy, Tonsillectomy, or Adenotonsillectomy

An adenoidectomy and a tonsillectomy are two of the most common surgeries particularly among children’s population. Those two common surgeries many times falsely considered as one procedure; however, they should be considered as two different ones. Depending on the reason, a tonsillectomy with an adenoidectomy may be indicated, especially in cases of SDB problems. They are performed when these structures are causing significant difficulties and conservative therapies are ineffective (Figure 3).

An intriguing review by Randall [14] delves into the current indications for otolaryngology consultation regarding tonsillectomies and adenoidectomies. The review is based on the comprehensive guidelines provided by the American Academy of Otolaryngology—Head and Neck Surgery (AAOHNS) in 2019, which were specifically tailored for children. Although these indications have been extended to adults in clinical practice, further research is necessary to validate their applicability.

The clinical indications for an adenoidectomy, as per the most recent guidelines from the AAOHNS, include the following: (1) recurrent acute otitis media (if a child has had three or more episodes in the past six months or four or more episodes within the past year); (2) chronic otitis media with effusion (if a child has persistent bilateral effusion for more than three months and associated hearing loss or other significant symptoms); (3) nasal obstruction and mouth breathing that affect a child’s quality of life, sleep patterns, or growth; (4) recurrent sinusitis (if a child has three or more episodes of acute sinusitis within a year, with each episode lasting for at least ten days and accompanied by documented sinusitis-related symptoms); and (5) facial growth abnormalities attributed to chronic adenoid hypertrophy, such as long face syndrome or an open bite.

The clinical indications for tonsillectomy, as stated by the AAOHNS, include the following: (1) recurrent tonsillitis (at least seven episodes in one year, five episodes per year for two consecutive years, or three episodes per year for three consecutive years); (2) recurrent peritonsillar abscess (if a patient experiences repeated episodes, despite appropriate medical treatment, tonsillectomy may be considered to prevent future abscesses and associated complications); (3) tonsillar asymmetry indicating malignancy, as well as cases of confirmed malignancy; and (4) obstructive SDB.

SDB refers to the recurrent obstruction of the upper airway during sleep, disrupting normal ventilation and sleep patterns. The symptoms include hyperactivity, daytime tiredness, and aggression, while the signs may include loud snoring, observed apnea, restless sleep, growth retardation, a poor school performance, and bedwetting. Children with SDB experience higher rates of antibiotic use, increased hospital visits by 40%, and a 215% rise in healthcare utilization due to them having more upper respiratory infections than those without SDB do [87]. Tonsillar and adenoid hypertrophy are the primary causes of SDB. However, the severity of SDB does not always correlate with the tonsillar size, and polysomnography may be used for further evaluation in patients exhibiting signs and symptoms of SDB, even without tonsillar hypertrophy. Indeed, there has been a rise in the prevalence of sleep apnea among young children (<3 years old) attributed to factors such as environmental pollution and an increased occurrence of allergic reactions. These cases of sleep apnea are often associated with the hypertrophy of the adenoids or tonsils.

Additional indications for tonsillectomy include halitosis, febrile seizure, certain syndromes like PFAPA syndrome, enuresis, dysphagia, dysphonia, dental malocclusion, and psoriasis [12].

### 6.4. Before and after Adenoidectomy, Tonsillectomy, or Adenotonsillectomy

Understanding the dynamic changes that occur before and after an adenoidectomy, tonsillectomy, or adenotonsillectomy is paramount for optimizing patient outcomes. Recent literature has provided valuable insights into these procedures, highlighting various aspects, such as immunological alterations, improvements in respiratory function and sleep, as well as enhancements in their quality of life and symptom relief.

Lacking tonsils eliminates the possibility of developing tonsillitis; however, individuals can still experience pharyngitis and sore throats. Although the palatine tonsils are believed to significantly contribute to the occurrence of chronic or recurrent acute throat infections, they are likely not the sole determining factor. Consequently, undergoing a tonsillectomy may help prevent future throat infections or lessen their severity, leading to notable improvements in daily functioning and the health-related quality of life for patients [88]. The immunological impact of tonsillectomy in children has been extensively discussed among healthcare professionals and is a concern for parents. Numerous studies have explored its effects on the immune system, resulting in diverse findings that have caused some confusion [89]. However, despite the variability in research, there is sufficient evidence to conclude that a tonsillectomy does not have clinically significant adverse effects on the immune system. A comprehensive follow-up study revealed that tonsillectomies do not compromise children’s immune functions. Over time, both the humoral and cellular immunity of patients recovered, exhibiting an immune capacity similar to that of healthy individuals of the same age. These results debunk the misconception that tonsillectomy jeopardizes life-long immunity [90]. To improve our understanding, future studies should employ standardized methodologies, including preoperative and control laboratory tests for comparison. Long-term follow-up assessments encompassing both humoral and cellular immunity would provide valuable perceptions. 

Accordingly, an adenoidectomy with or without a tonsillectomy may briefly lower serum IgA levels in young children, but this is usually temporary and does not increase the risk of infection or immune-deficiency disorders in children under 3 years old. The presence of remaining mucosa-associated lymphoid tissue suggests compensation for the absence of adenoid and tonsil tissue [91]. Moreover, the adenoids and tonsils harbor a diverse microbial community, and their removal or reduction via surgery can have a profound effect on the composition of the microbiome.

Obstructive sleep apnea (OSA) is a common indication for these surgical interventions, and postoperative improvements in respiratory function, snoring reduction, and the resolution of OSA symptoms have been well documented. Recent literature further emphasizes the effectiveness of these procedures in improving sleep quality, oxygen saturation levels, and daytime cognitive functioning in both pediatric and adult populations [92]. Furthermore, adenoidectomies, tonsillectomies, and adenotonsillectomies have been associated with significant improvements in the quality of life for individuals suffering from chronic infections, sleep disturbances, and related symptoms. The latest studies demonstrate enhanced well-being, reduced sick days, and improved cognitive performance following these surgical interventions among both children and adults [93,94,95]. These procedures have also shown to provide symptom relief in conditions such as chronic tonsillitis, snoring, SDB, and associated comorbidities. 

By integrating the findings from recent literature into clinical practice, we can optimize patient care and tailor management strategies to achieve better outcomes for individuals undergoing these surgical interventions.

## 7. Conclusions

The adenoids and tonsils represent vital constituents of the immune system, carrying profound implications for immunology, physiology, and the microbiome. Through an examination of cutting-edge research and advancements, we present valuable insights into the potential impact of these structures on overall healthy and diseased states. Additionally, we explored the evolving role of surgical interventions in the management of adenoids and tonsils, providing a comprehensive understanding of the associated benefits and challenges. By establishing robust groundwork for future investigations, this review seeks to foster innovative approaches in comprehending and addressing the intricate interplay between adenoids, tonsils, and various aspects of human health. Through this comprehensive understanding, we pave the way for a more precise and effective care pathway for individuals facing these intricate challenges.

## Figures and Tables

**Figure 1 microorganisms-11-01624-f001:**
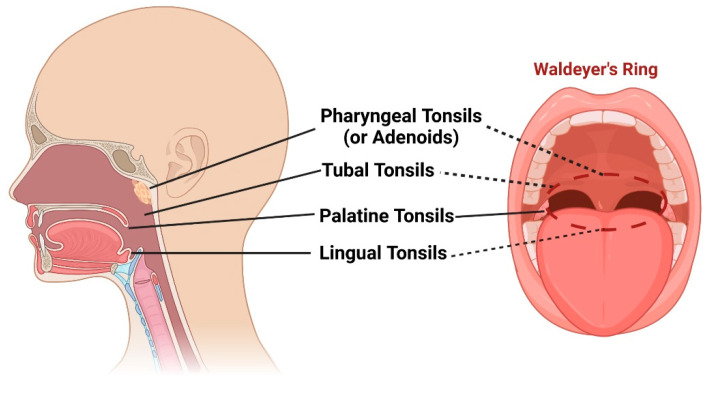
Schematic representation of pharyngeal tonsils (or adenoids), tubal tonsils, palatine tonsils, and lingual tonsils, as well as the “intelligible ring” that they form (Waldeyer’s ring). Created with BioRender.com (accessed on 29 May 2023).

**Figure 2 microorganisms-11-01624-f002:**
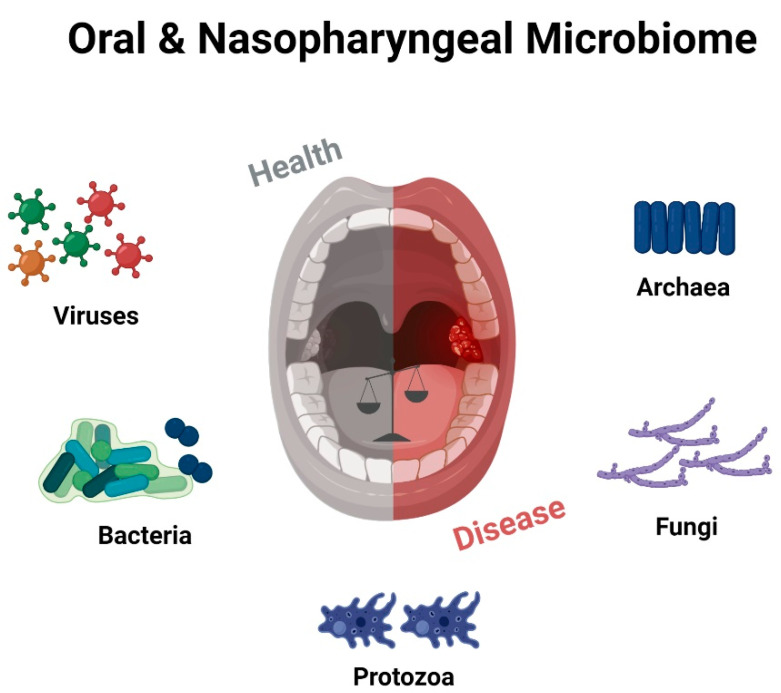
Schematic representation of the numerous microbial communities inhabiting the oral and oropharyngeal cavity, highlighting the intricate “equilibrium” between healthy and diseased states. Created with BioRender.com (accessed on 29 May 2023).

**Figure 3 microorganisms-11-01624-f003:**
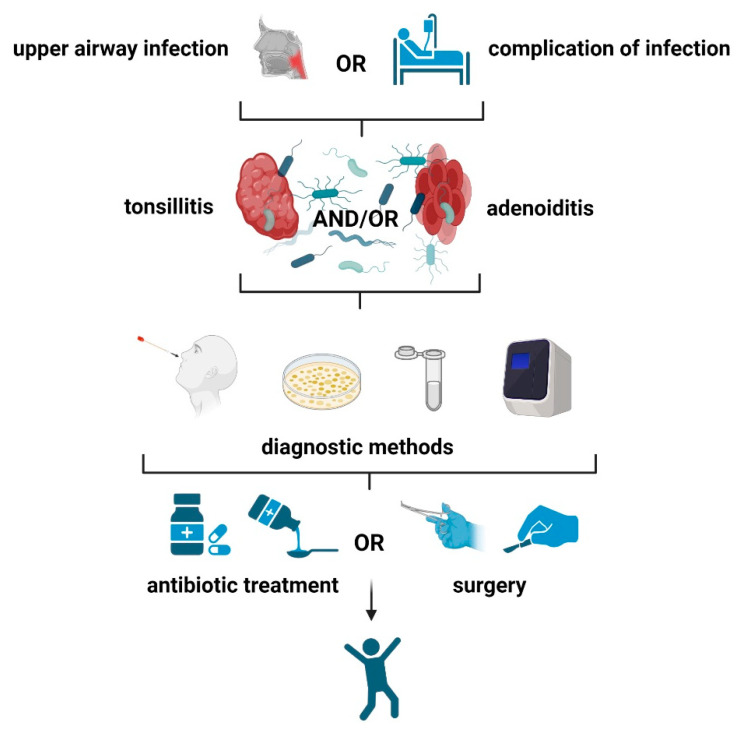
This “algorithm” outlines the general steps (diagnosis-conservative therapy-surgical procedures) to be followed in cases of upper respiratory infections (tonsillitis, adenoiditis, or adenotonsillitis) for appropriate pathogen treatment. Created with BioRender.com (accessed on 29 May 2023).

## Data Availability

Not applicable.

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
