# Peer review of "Unveiling the Enigmatic Adenoids and Tonsils: Exploring Immunology, Physiology, Microbiome Dynamics, and the Transformative Power of Surgery"

_microorganisms, 2023, doi:10.3390/microorganisms11071624_

Round 1

Reviewer 1 Report

The microbiome is presented in detail and up-to-date. I would like to add information about the immunological component, especially in patients with abscess or even more so with repeated peritonsillar abscesses. what are the recommendations for rehabilitation after surgery? What information is there on rehabilitation after surgery ?

Author Response

First of all, we would like to express our gratitude for the constructive comments that have greatly improved our manuscript. We agree with the reviewer's comment and have incorporated additional information with corresponding references into the text (lines 154-172 and lines 320-352). In each section highlighted by the reviewer, we have provided relevant literature to support our statements. We have discussed what is currently known about the immunological aspects of the peritonsillar abscess, its management, potential risk factors, etc. Thank you for pointing out the areas that needed further development and for providing us with the opportunity to enhance the manuscript.

Reviewer 2 Report

Pinelopi Samara et al. manuscript entitled “Unveiling the enigmatic adenoids and tonsils: exploring immunology, physiology, microbiome dynamics, and the trans- 3 formative power of surgery” is a review showing the importance of the mucosal immune system of the Waldeyer's ring, anatomical zone of the oro-rino- pharyngeal district constituted by lymphatic tissues between the palatine tonsils and adenoids. It Is a very interesting work.

Only minor revisions for the authors:

Arguments better also the interest in viruses, which is little described in the review. I recommend that these papers be well evaluated as well:

1. Zajac, V.; Mego, M.; Martinický, D.; Stevurková, V.; Cierniková, S.; Ujházy, E.; Gajdosík, A.; Gajdosíková, A. Testing of bacteria isolated from HIV/AIDS patients in experimental models. Neuro Endocrinol. Lett. 2006, 27 (Suppl. S2), 61–64. [Google Scholar])

2. Zajac V, Matelova L, Liskova A, Mego M, Holec V, Adamcikova Z, Stevurkova V, Shahum A, Krcmery V. Confirmation of HIV-like sequences in respiratory tract bacteria of Cambodian and Kenyan HIV-positive pediatric patients. Med Sci Monit. 2011 Feb 25;17(3):CR154-8. doi: 10.12659/msm.881449. PMID: 21358602; PMCID: PMC3524724.

Author Response

First of all, we would like to express our gratitude for the constructive comments that have greatly improved our manuscript. We agree with the reviewer's comment and have incorporated additional information with corresponding references into the text (lines 237-260). The reviewer was absolutely right in pointing out that we did not extensively address the virus part. As a result, we have incorporated text describing the viruses detected in the tonsils and adenoids of asymptomatic children. Furthermore, we have included a specific group of patients with HIV, as suggested in the second article that focused on the respiratory tract, and included relevant and valuable data. We greatly appreciate both the first article, which provided us with valuable insights (although not directly incorporated into the text as it pertained to the gastrointestinal tract), and the second article that prompted us to enhance the discussion on viruses and HIV infection. Thank you sincerely for your valuable feedback.